# Beyond Protected Attributes: Disciplined Detection of Systematic Deviations in Data

**Adebayo Oshingbesan**
IBM Research Africa
adebayo.oshingbesan1@ibm.com

**Winslow Georgos Omondi**
IBM Research Africa
omondiwinsly2@gmail.com

**Girmaw Abebe Tadesse**
IBM Research Africa
girmaw.abebe.tadesse@ibm.com

**Celia Cintas**
IBM Research Africa
celia.cintas@ibm.com

**Skyler Speakman**
IBM Research Africa
skyler@ke.ibm.com

## Abstract

Finding systematic deviations of an outcome of interest in data and models is an important goal of trustworthy and socially responsible AI. To understand systematic deviations at a subgroup level, it is important to look beyond *predefined* groups and consider all possible subgroups for analysis. Of course this exhaustive enumeration is not possible and there needs to be a balance of exploratory and confirmation analysis in socially-responsible AI. In this paper we compare recently proposed methods for detecting systematic deviations in an outcome of interest at the subgroup level across three socially-relevant data sets. Furthermore, we show the importance of looking through all possible subgroups for systematic deviations by comparing detected patterns using only protected attributes against patterns detected using the entire search space. One interesting pattern found in the OULAD dataset is that while having a high course load and not being from the highest socio-economic decile of UK regions makes students 2.3 times more likely to fail or withdraw from courses, being from Ireland or Wales mitigates this risk by 37%. This pattern may have been missed if we focused our analysis on the protected groups of gender and disability only. Python code for all methods, including the most recently proposed "AutoStrat" are available on open-sourced code repositories.

## 1  Introduction

Understanding data at a subgroup level is an important part of building trustworthy, reliable, and socially responsible AI-based systems [1, 2]. Systematic deviations in data can help reveal important issues such as predictive bias [3], distribution shifts [4], and under/over representation in data [5, 6], among others. Most of the works in Trustworthy AI have involved analysis of unfair or unequal treatment of an individual (or group) based on certain characteristics such as gender, race, marital status, etc., that should not be used for making decisions, usually known as protected attributes [7, 8]. However, focusing on some predefined attributes for systematic deviations can lead to fairness gerrymandering where relevant and important patterns that do not appear at a group or individual level become visible at the subgroup level [9, 10]. In a dataset with binary definitions of race and gender, outcomes assigned to black men or white women independently and uniformly at random in a population will appear equitable when one considers the protected groups of gender and race separately but is actually inequitable to black women, and white men subgroups in the data [9].

While it might be possible to enumerate all the possible subgroups from a handful of pre-determined protected attributes, we could still be missing out on more relevant and important patterns. For

2022 Trustworthy and Socially Responsible Machine Learning (TSRML 2022) co-located with NeurIPS 2022.

example, outcomes could be assigned such that it would be fair to the four possible protected subgroups involving race and gender but may be unfair to a subgroup such as young married people or black women without a college degree when we add the unprotected attributes of age and education level. We need to consider all the possible subgroups in the data to advance towards a more trustworthy and responsible analysis. However, there are three major obstacles to performing the analysis. First, there are exponentially-many subgroups to consider, which means we cannot perform a brute-force search. This is why many analyses focus on protected groups only, as it makes the discovery process more tractable. Furthermore, a heuristic-based search, such as a beam search, might miss out on important patterns. Second, since there are a lot of subgroups and, by extension, hypotheses to test, we may run into the problem of a high false discovery rate due to multiple hypothesis testing [11, 12]. Third, given several subgroups with statistically significant deviations, there is still the important question of how to rank them correctly (see Appendix A). To address these concerns, we use a technique that leverages mathematical properties of commonly used methods of divergence to perform this search using standard compute resources [13, 14].

In summary, our contributions are as follows:

- We describe Automatic Stratification (AutoStrat) - an efficient algorithm to automatically **discover** systematic deviations in data in a more trustworthy and responsible way by going beyond protected subgroups.
- We discuss the notion of divergence of a subgroup that automatically balances the size of the subgroup and the extremity of its systematic deviation.
- We show the importance of disciplined subgroup discovery for three use cases - under/over representation in data and model risk assessment.
- We finally compare the subgroups discovered by our approach to the subgroups discovered in other subgroup discovery algorithms, showing the utility of allowing for more than one feature value per feature in the subgroup description.

## 2 Related Work

There have been several attempts to understand interactions present in data used for machine learning purposes [15, 16, 17]. However, this analysis is usually limited to two or three features, and it requires domain knowledge which makes it difficult to identify critical but previously unknown sub-populations. Other works like [18, 19] have focused on the explainability of black-box models at an instance or global level of the data in order to uncover bias in how the model works or understand why the model is making certain predictions. On the other hand, feature selection methods [20, 21] try to identify relevant features for prediction tasks to help the model generalize better to unseen data. More recent works [22, 23, 24] have tried to detect when unseen data deviates significantly from the training set. While all these methods are useful, they do not answer the critical question of how do we automatically detect and characterize systematic deviations in data at a subgroup level for the exponentially many subgroups.

Subgroup discovery aims at automatically extracting subgroups with much higher (or lower) rates of the target outcome in a given data [25, 26]. Pysubgroup [26] is an open-source implementation of several subgroup discovery algorithms where subgroups are a conjunction of simple predicates (logical ANDs only). Two recently proposed model validation algorithms, Slice Finder [27] and DivExplorer [28], could be applied to the subgroup discovery task(see Appendix B). Table 1 shows an overview comparison of these methods with our approach, AutoStrat. Unlike the existing methods, which only support logical AND combinations of feature values, AutoStrat also supports logical OR combinations, resulting in a more flexible description of the identified subset (see Appendix C). Finally, AutoStrat optimises for the mathematically correct balance of both the size and severity of the deviation of identified subgroup, thereby reducing the likelihood of false discoveries.

## 3 Approach: AutoStrat

Multi-dimensional Subset Scan (MDScan) was originally proposed in a spatial-temporal setting with applications in bio-surveillance and situational awareness [13, 29]. More recently, MDScan was re-framed on tabular data and used to detect predictive bias in black box classifiers [30]. Our current

Table 1: Comparison of Automatic Subgroup Discovery Methods

| | Supports Logical ORs | Auto-balances significance & size | Subgroup vs Full data | Task Agnostic | Uses heuristics |
|---|---|---|---|---|---|
| **Pysubgroup** [26] | | ✓ | ✓ | ✓ | ✓ |
| **Slice Finder**[27] | | | | ✓ | ✓ |
| **Div-Explorer** [28] | | | ✓ | | ✓ |
| **AutoStrat (Ours)** | ✓ | ✓ | ✓ | ✓ | |

work, AutoStrat, builds on top of MDScan with tabular data but without the focus on predictive bias in classifiers. Rather, our goal is to identify anomalous subgroups that have higher-than-average outcomes as compared to the global mean.

Given a dataset $D$ and a search space $\lambda$, we describe a subgroup, $S$ as a conjuction of literals such that we have *logical ANDs* at the feature level and *logical ORs* at the feature value level for features in $\lambda$, *e.g., Country = {Kenya OR Nigeria} AND Marital Status = {Single OR Married}*. This can also be referred to as a subspace of the Cartesian product of categorical features and their values. If $\lambda$ is limited to just the protected features, then the subgroup discovered is called a protected subgroup (PS). Otherwise, if $\lambda$ includes features beyond the protected attributes then the anomalous subgroup is a Beyond-protected subgroup (BPS).

MDScan is an iterative ascent procedure (see Algorithm 1 for pseudocode) where each step is efficient and exact due to the Additive Linear-time Subset Scanning property (ALTSS) [31, 32]. While a feature containing $k$ unique values has exponentially many ($O(2^k)$) subgroups, a scoring function that satisfies the ALTSS property guarantees that the most anomalous (highest scoring) subgroup will be one of only linearly-many ($O(2K)$) subgroups. This property drastically reduces the search space and makes logical ORs between feature values tractable. Using this property, Bias Scan [30] searches for a subset $S$ that maximizes the divergence between the observed and expected outcomes in $S$ where $y_i$ is a binary outcome of interest for record $i$ and $e_i$ is probability assigned to the outcome of interest for record $i$ by a predictive model.

While Bias Scan focuses on identifying significant predictive bias in classifiers, AutoStrat pivots away from predictive models by replacing $e_i$ with the proportion of the target class $y$. This enables us to perform subgroup discovery by maximizing Equation 1. Given a subgroup $S$, the scoring function, $\Gamma(S)$, is

$$\Gamma(S) = \sum_{i \in S} (y_i \cdot \log{(q)} - \log{(1 - e_i + q \cdot e_i)}), \tag{1}$$

Where $y_i$ is observed outcome for the $i^{th}$ record, $e_i$ is the expected outcome and the proportion of the target class in $y$, and q is the multiplicative odds increase of the subgroup, $S$. This is also referred to as the "Bias score" of a subset [30]. This scoring function can be derived from a likelihood ratio statistic based on two Bernoulli distributions (see [33]). The denominator (null hypothesis) is that the odds of outcome $y_i$ are $\frac{e_i}{1-e_i}$ for all $i$. The numerator (alternative hypothesis) is that the odds of $y_i$ have been increased by a multiplicative factor, $q$ for records in some subset, $S$. By maximizing this statistic over all subsets, we are finding the subset that shows the most evidence that $q > 1$.

**Significance testing** To ascertain the statistical significance of the found subgroup, randomization-based hypothesis testing is used [34, 35]. We use randomization-based hypothesis testing because of the exponential nature of the number of the hypothesis we are testing during the discovery process. Correction mechanisms such as the Benferroni correction [36] and the Benjamini-Hochberg correction [37] are too conservative and would incorrectly flag many subgroups discovered as false discoveries. To carry out randomization-based hypothesis testing, we consider $K = 100$ iterations, and a replica of the observed outcome is generated from the assumption of the null hypothesis for each iteration, $k$. Scanning for the anomalous subset is applied to each replica of the outcome and the divergence scores from all the replicas ($\Gamma(S_k)$, $k = 1, 2, \cdots, 100$) are compared with the true divergence score ($\Gamma(S^*)$).

An empirical $p$-value ($p$) is computed as $p = (r(S^*) + 1)/(K + 1)$, where $r(S^*) = \sum_{k=1}^{K} \zeta_k(S^*)$, and $\zeta_k(S^*) = 1$ if $\Gamma(S_k) \geq \Gamma(S^*)$ else $\zeta_k(S^*) = 0$.

## 4 Experimental Setup

**Datasets** We use three commonly used datasets for Trustworthy AI research - Compas [3], Credit card clients [38], and OULAD education data [39], covering three important use cases - under/over representation in data and model risk assessment. The analysis for the OULAD dataset focuses on under representation of pass/distinction in the data while that of the Credit card client dataset focus on the over representation of default payment in the next month. For the Compas dataset, the outcome is the risk of violence as assigned by the Compas tool. By focusing on a predictive model's predictions without *the ground truth*, we are performing a model risk assessment [19, 40] to check the consistency of the model across subgroups rather than how it performs in relation to the ground truth (model validation) or why it makes its predictions (model explainability). Table 2 provides an overview of each dataset, including the number of records, search space size, protected attributes, etc. A full description of the datasets and their features can be found in [41].

**Baseline methods** We compare AutoStrat against four other recently proposed subgroup discovery algorithms - Beam Search and Apriori (as implemented by Pysubgroup [26]), Fp-growth algorithm (as implemented by DivExplorer [28]), and Slice Finder [27]. More details on these methods are provided in the appendix.

**Evaluation metrics** We evaluate the anomalousness of an identified subgroup ($S$) using the Bernoulli likelihood ratio statistic, $\Gamma(S)$. Given $p_1$ as the proportion of outcomes in the records belonging to $S$, $p$ as the proportion of outcomes in the entire dataset, $n'$ as the number of records in $S$, $n$ as the total of records in the dataset, and $p_0$ as proportion of outcomes in the records not belonging to $S$, we report four other statistics - Lift 2, Support 3, Odds Ratio ($OR$) 4, and Weighted Relative Accuracy ($\phi(S)$) 5 in addition to the anomalous score, $\Gamma(S)$. While Lift tells us how much more likely the target outcome in a subgroup is in comparison to the average, Support tells us the relative size of the subgroup to the entire data. The odds ratio and weighted relative accuracy both measure how extreme the deviation of the subgroup is. Finally, we include runtimes for all the experiments in seconds. Code for all the experiments and analysis can be found at `https://ibm.biz/BPS-Autostrat`.

$$Lift = \frac{p_1}{p} \tag{2}$$

$$Support = \frac{n'}{n} \tag{3}$$

$$OR = \frac{p_1/(1 - p_1)}{p_0/(1 - p_0)} \tag{4}$$

$$\phi = \left(\frac{n'}{n}\right)(p_1 - p) \tag{5}$$

## 5 Preliminary Results and Discussion

### 5.1 Comparison between protected subgroups and non-protected subgroups

We ran experiments to obtain the subgroup with the most systematic deviation over the entire search space (Beyond Protected Subgroups, BPS) from AutoStrat and over only the protected attributes (PS) as defined in Table 2. We also report the empirical $p$-values as obtained using randomization-based significance testing 3. Table 3 presents the results of the comparison between protected subgroups and beyond protected subgroups across the three datasets. As reported across several research [3, 41], the patterns found in protected attributes are statistically significant ($p$-value$< 0.05$) for both the Compas and OULAD datasets but not significant for the Credit card dataset. While the *age_cat is less than* 25 subgroup could have been found by other techniques (e.g. a logistic regression analysis

Table 2: Details of datasets used for validation

| Dataset | Number of Records | Protected Attributes | Possible Subgroups (Without logical ORs) | Target | Outcome Proportion |
|---------|-------------------|----------------------|------------------------------------------|--------|--------------------|
| Compas | 4,743 | Sex, Race | 250,047 (432) | v_decile_score >5 | 0.2043 |
| Credit Card | 30,000 | Sex, Education, Marriage | 2.79E+62 (3.06E+21) | default payment next month | 0.2212 |
| OULAD | 32,593 | Gender, Disability | 3.12E+13 (218,400) | final results = pass or distinction | 0.3151 |

Table 3: Comparison of the identified protected subgroup (PS) and Beyond-protected subgroup (BPS) across different datasets using odds ratio ($OR$), Bernoulli likelihood statistic ($\Gamma(S)$), and empirical $p$-value.

| Dataset | Type | Subgroup | $p$-value | $OR$ | $\Gamma(S)$ |
|---------|------|----------|-----------|------|-------------|
| Compas | BPS | age_cat = Less than 25 | 0.0099 | **9.33** | **274** |
| | PS | sex = Male AND race = African-American OR Native American | 0.0099 | 1.86 | 70 |
| Credit Card | BPS | PAY_0 = 2 OR 3 OR 4 | 0.0099 | **11.55** | **1583** |
| | PS | MARRIAGE = 1 OR 2 OR 3 AND SEX = 1 AND EDUCATION = 2 OR 3 | 0.5445 | 1.27 | 40 |
| OULAD | BPS | studied_credits = 90.0 - 655.0 AND region = NOT (IRELAND or WALES) AND imb_band = 0%-90% | 0.0099 | **2.26** | **309** |
| | PS | disability = Y | 0.0099 | 1.42 | 43 |

on a one-hot encoded version of the dataset [3]) because of its relative simplicity, this significant pattern would have been missed if we only focused on protected attributes. In the OULAD dataset, we see that while the protected subgroup has a significant deviation, when we maximize Eq. 1 over all features beyond just the protected ones, we uncover a socially relevant group: students who are not from the highest socio-economic decile regions of the UK excluding Ireland and Wales that study between 90 - 655 credits units. This group shows the most evidence that pass or distinction rates are lower than average. It is interesting that while not being from the highest socio-economic decile regions of the UK coupled with a high course workload makes a student more likely to fail or withdraw from courses, students from Ireland and Wales seem to be protected against this pattern as the odds ratio drops by 33% from 2.26 to 1.42. Similarly, while focusing on *just* protected attributes may have given investigators a false sense of security regarding the extent of systematic deviations in the Credit card dataset, AutoStrat showed that a high systematic deviation exists in this data through an efficient search of an exponentially large space.

## 5.2 Comparison with other recently proposed subgroup discovery algorithms

Following the experimental results from section 5.1 that showed the importance of not limiting our search space to only some pre-determined protected attributes, we ran experiments to extract beyond protected subgroups from the baseline methods (see section 4) and then compare the metrics of the extracted subgroups from these algorithms to the extracted subgroups from AutoStrat. Table 4 presents the results of the comparative analysis of AutoStrat when compared to these other subgroup discovery algorithms. There are three key takeaways from these results. First, AutoStrat is able to consistently find deviations that are the largest across all the datasets, whether that deviation is measured by the AutoStrat score, the Weighted Relative Accuracy, or the Odds Ratio (with one exception). This exception occurred because AutoStrat correctly balances the fact that 20% of the

Table 4: Comparisons of subgroup discovery algorithms across three datasets using Lift, Support, Odds Ratio ($OR$), Bernoulli Likelihood Statistic ($\Gamma(S)$), and Weighted Relative Accuracy ($\phi(S)$)

| Method | Subgroup | Lift | Support | $OR$ | $\Gamma(S)$ | $\phi(S)$ | Time |
|---|---|---|---|---|---|---|---|
| **Compas** | | | | | | | |
| AutoStrat (Ours) | age_cat = Less than 25 | 2.77 | 0.18 | **9.33** | **274** | **0.07** | 3.03 |
| Beam search [26] | age_cat = Less than 25 | 2.77 | 0.18 | **9.33** | **274** | **0.07** | **0.01** |
| Apriori [26] | age_cat = Less than 25 | 2.77 | 0.18 | **9.33** | **274** | **0.07** | 1.30 |
| Fp-growth [28] | race = African-American AND age_cat = Less than 25 | 3.07 | 0.1 | 9.17 | 209 | 0.04 | 0.37 |
| Slice Finder [27] | age_cat = Less than 25 | 2.77 | 0.18 | **9.33** | **274** | **0.07** | 3.48 |
| **Credit Card** | | | | | | | |
| AutoStrat (Ours) | PAY_0 = 2 OR 3 OR 4 | 3.16 | 0.1 | **11.55** | **1583** | **0.05** | 13.28 |
| Beam search [26] | PAY_2 = 2 | 2.51 | 0.13 | 6.09 | 1034 | 1.57 | **0.4** |
| Apriori [26] | PAY_2 = 2 | 2.51 | 0.13 | 6.09 | 1034 | 0.04 | 1.82 |
| Fp-growth [28] | PAY_2 = 2 | 2.51 | 0.13 | 6.09 | 1034 | 0.04 | 11.81 |
| Slice Finder [27] | PAY_2 = 2 | 2.51 | 0.13 | 6.09 | 1034 | 0.04 | 13.44 |
| **OULAD** | | | | | | | |
| AutoStrat (Ours) | imd_band=0% - 90% AND studied_credits = 90 - 120 OR 120 - 655 AND region = NOT(Ireland OR Wales) | 1.47 | 0.2 | 2.26 | **309** | **0.03** | 25.44 |
| Beam search [26] | studied_credits=90 - 120 | 1.29 | 0.2 | 1.67 | 118 | 0.02 | **0.72** |
| Apriori [26] | studied_credits=90 - 120 | 1.29 | 0.2 | 1.67 | 118 | 0.02 | 6.72 |
| Fp-growth [28] | num_prev_attempts=0 AND studied_credits=90 - 120 | 1.31 | 0.16 | 1.68 | 109 | 0.02 | 11.73 |
| Slice Finder [27] | region=North Western AND num_prev_attempts=2 | 1.85 | 1.91E-3 | **3.05** | 9 | 0 | 260.77 |

dataset having an odds ratio of 2.26 is a bigger systematic deviation than 0.191% of the dataset having an odd ratio of 3.05 in the OULAD dataset. Second, AutoStrat is able to find subsets that the alternative methods would not find because of its ability to support more than one feature value per feature (logical ORs) in its subgroup description. This is effectively shown in the results for the Credit card dataset, where the other methods all find the same pretty significant deviation ($PAY\_2 = 2$) but missed a significantly larger deviation that had almost 2x the odds ratio because they did not take into account the OR relationships ($PAY\_0 = 2$ OR $3$ OR $4$). Finally, despite searching through a much bigger search space (see Table 2), AutoStrat is still competitive in terms of runtime[1]. Unsurprisingly, the heuristic beam search is the fastest. However, it has to be noted that it is not guaranteed to find the largest deviations.

---

[1]*Given heterogeneous runtimes across methods when using the larger dataset, we report the runtime for Slice Finder based on 10% of the data.

# 6 Conclusion

In this paper, we described AutoStrat - an efficient algorithm for divergent subgroup discovery in a way that correctly balances the significance of the divergence and the size of the subgroup while supporting the addition of more than one feature value for a feature in the subgroup description. We then showed how this approach enables us to find more divergent subgroups across different tasks, some of which we would have missed out on if we only focused on subgroups involving the protected groups. We also compared the results discovered by the described algorithm with other recently proposed algorithms. One limitation of AutoStrat, like the other subgroup discovery algorithms, is the need to bin continuous features. Future works include supporting continuous variables directly and applying this approach beyond tabular datasets to unstructured data such as images and text. Also, while we only focused on the most divergent subgroup in this paper, we would be extending the analysis to include multiple subgroups in future works.

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

## Appendix A  How Size and Extremity of Deviation Trade-off Affects Ranking

Consider five subgroups - A, B, C, D, and E as described in Table 5. To rank these subgroups, we have to take into account their relative sizes and the extremity of their deviations (i.e. the proportion of target outcome present in the subgroup relative to its size). Subgroup A contains 20% of the data and 30% of the target outcome. Subgroup B contains 5% of the data and 15% of the target outcome. The relative difference between the proportion of target outcome and relative size is equal for these subgroups, however this does not mean they are equally anomalous because when one takes a look at the relative ratio rather than the relative difference, subgroup B seems to be more extreme. Similarly, subgroup D has a higher relative difference and larger size when compared to subgroups B and C but they both have a higher relative ratio. Furthermore, subgroup E has a much higher relative ratio than the other subgroups but it is a very small proportion of the data with a small relative difference. It is not directly clear how to balance all these different factors to rank these subgroups. Thus, there must be a correct mathematical balance between the size of the found subgroup and the extremity of its deviation to correctly rank these subgroups.

Table 5: Table of Properties of Hypothetical Subgroups

|   | Relative Size | Proportion of Target Outcome |
|---|---|---|
| A | 0.2 | 0.3 |
| B | 0.05 | 0.15 |
| C | 0.01 | 0.02 |
| D | 0.3 | .5 |
| E | 0.001 | 0.01 |

# Appendix B    Baselines

## B.1    Pysubgroup

Pysubgroup implements several subgroup discovery search mechanism such as Beam Search, Apriori, Depth First Search, Best First Search, among others. There are also several task-specific quality functions also implemented. While it automatically balances the size and extent of divergence through its quality score, the user is still required to pass in a depth parameter that controls how many possible logical ANDs can exist in the returned subgroups. However, as shown in table 1, it does not support logical ORs. In this analysis, the depth parameter was set at 5 and the quality function was weighted relative accuracy (Eq. 5).

While Eq. 5 does try to automatically balance size and significance of the systematic deviation, it is limited by the fact that the simple difference in proportion of outcomes in the subgroup and entire population does not fully correctly capture the deviation. For example, consider two subgroups from two datasets with a relative size of 20% but for subgroup 1, the subgroup proportion is 25% vs 5% in the entire dataset while for subgroup 2, the subgroup proportion is 40% vs 25% in the entire dataset. Eq. 5 would assign these two subgroups the same score despite the fact that subgroup 1 had a 3x increase in odds ratio vs 2x increase in subgroup 2.

## B.2    DivExplorer

DivExplorer defines subgroups as a conjuction of logical ANDs. The quality score measures differences in metrics such as false positive rate, false negative rate, error rate, and accuracy for subgroup the entire dataset. Candidates are generated using frequent pattern mining techniques to identify itemsets and thus is focused only on classification tasks. Continuous values are also required to be binned for candidates to be generated. It does not adopt early stopping heuristics of stopping search but has a post-processing pruning stage where items whose marginal contribution to divergence of an itemset is lower than some threshold are pruned. User must define a minimum Support threshold as it does not automatically balance significance and size of divergence. In this analysis, the minimum Support was set as 0.1 and the metric was accuracy. DivExplorer's accuracy metric is an unweighted of Eq. 5 (i.e. we do not multiply by the relative size of the subgroup). Thus, DivExplorer will return subgroups where the relative proportion in the subgroup is much higher than the average given that the subgroup has a relative size greater than the minimum Support threshold.

## B.3    Slice Finder

Slice finder focuses on more interpretable subsets i.e subsets with fewer number of literals. These slices, are a conjuction of feature-value pairs (i.e logical ANDS only). Continuous values are expected to be discretized to help numeric feature values to be more sizable and meaningful. Slice finder uses an exhaustive lattice search algorithm in breadth-first mode to search for interesting slices, terminating search when either k significant slices are found or there are no more slices to explore. The quality function is an effect size that test the extent of the differences in the loss distributions of a slice and its complement where complement is all records in the dataset that is not in the subgroup(Eq. 6. For a classification task, the loss function in Eq. 6 is the logloss and the difference is the KL-divergence. The effect size during subgroup discovery measures the how much the distribution of outcomes in the subgroup differs from the mean proportion in the dataset. The effect size is also used as an early stopping and post-processing heuristic. Thus, the user has to pass in the effect size as a parameter. Additionally, a degree parameter controls the maximum logical ANDs the returned slice can contain is also required. In this analysis, the effect size and degree were 0.5 and 5 respectively. Because of runtime issues, only 10% of each dataset was used for slice finder.

$$\sqrt{2} * \frac{\psi(S, h) - \psi(S', h)}{\sqrt{\sigma_s^2 + \sigma_{s'}^2}} \tag{6}$$

Where:

$\psi(S, h) - \psi(S', h)$ is difference in loss function between the subgroup and its complement

$\sigma_s^2, \sigma_{s'}^2$ are variances of losses in subgroup & its complement respectively

## Appendix C   Importance of Logical ORs

Supporting logical OR combinations ensures that a subgroup discovery method does not miss important subgroups that could be excluded because of size or extremity of deviation. For example, given two subgroups, A and B, with relative sizes of 7% and 5% and descriptions red cats and black cats respectively, these subgroups will be excluded completely in the discovery process if the minimum relative size is set to 10% (DivExplorer requires a minimum relative size to be set) or might not have a deviation high enough to be considered (Slice Finder requires minimum effect size to be set). However, a subgroup C that is equivalent to the combination of subgroup A and B through the use of Logical ORs (i.e red or black cats) might be an important subgroup with a high relative size and extremity. This type of subset would be missed completely even if multiple rules are returned and then post-processed to include Logical OR combinations as they would not have been returned as interesting subgroups. It is important that Logical OR combinations are a part of the discovery process for these types of subgroups. Similarly, Logical OR combinations generally give more sizeable subgroups which ensures that we find subgroups that are not just statistically significant but are also meaningfully substantial as they are large enough to be considered important for real world use.

## Appendix D   MDScan Pseudo-code

---

**Algorithm 1:** Pseudo-code for Multi-dimensional subset scanning (MDScan)

---

1  *# Input and output definition*;
   **input**   :Dataset: $\mathcal{D} = \{(x_i, y_i)|i = 1, 2, \cdots, N\}$, Set of features:
          $\mathcal{F} = [f_1, f_2, \cdots, f_m, \cdots, f_M]$
   **output**:$AnomSubset$,
          $AnomScore$

2  *# Initialization* ;
3  $AnomSubset \leftarrow \{\}$;
4  $AnomScore \leftarrow -\infty$;
5  $UnCheckedF \leftarrow \mathcal{F}$;
6  *#Iterate until convergence*;
7  **while** $UnCheckedF$ *isNot* $\{\}$ **do**
8     *# Randomly select unchecked feature*;
9     $f_m \leftarrow$ Random($UnCheckedF$);

10    *# Mark the feature as checked*;
11    $UnCheckedF \leftarrow UnCheckedF \setminus f_m$;
12    *# Compute the anomalous score*;
13    $Score, Subset \leftarrow$ ALTSS ($f_m|AnomSubset$);

14    *# Compare the new score with previous best*;
15    **if** $Score > AnomScore$ **then**
16       *# Update the score, subset and reset the flag to unchecked*;
17       $AnomScore \leftarrow Score$ ;
18       $AnomSubset \leftarrow Subset$ ;
19       $UnCheckedF \leftarrow \mathcal{F}$;
20    **else**
21       Go to Step 4 ;

22 *# Return the most anomalous score and its subset*;
23 **return** $AnomSubset$, $AnomScore$

---

