# OpenReview forum: "Beyond Protected Attributes: Disciplined Detection of Systematic Deviations in Data"
_NeurIPS.cc/2022/Workshop/TSRML — TSRML2022_

### Official Review · Reviewer_oc7D · 2022-10-20
**Good paper, accept**

**Overall Rating:** 7

**Summary:**

This work proposes AutoStrat, an algorithm for discovering subgroups with high anomality or prediction bias. It can tackle the fairness gerrymandering problem, and can find attributes other than the protected ones on which the model has a bias. It is based on the MDScan method, which with the ALTSS assumption can find the subgroup with the highest anomality in linear time. The proposed method is compared with previous methods on real tabular datasets, and the results show that it achieves competitive performances.

**Strengths:**

Significance: The difficulty of finding subgroups with potential bias is a well known problem in this field, but there is little work that addresses it. Some methods like DRO and CVaR can train fair models without identifying these subgroups, but these methods usually achieve much lower performances than group-aware methods like group DRO. Therefore, discovering these subgroups is an important problem, and this work provides a nice approach.

Clarity: The paper is very well written and easy to follow. The method it proposes is intuitive, and the evaluation metrics the authors use to assess the methods make sense.

Novelty: Currently there is very little work addressing this problem. While the proposed algorithm is based on previous methods (MDScan), it fits this problem very well and thus should be considered novel.

**Weaknesses:**

1. I suggest the authors elaborate more on the intuition behind Equation (1). It looks like the log likelihood of something, and the authors can write that derivation out explicitly to make it clearer to the readers.
2. I also suggest the authors elaborate more on the evaluation metrics (2)-(5). For example, what is the intuition behind these metrics? Why is a higher OR better? These metrics do make sense, but a more detailed explanation can make them more compelling.
3. Currently the algorithm only works for tabular data with well defined attributes. For more general data, is it possible to use the algorithm too? For example, in combination with clustering.

**Overall Recommendation:**

Overall, I think this paper addresses a very important yet understudied problem, and proposes a very intuitive approach with good results. Thus, I recommend acceptance.

**Review Confidence:**

4: The reviewer is confident but not absolutely certain that the evaluation is correct

---

### Official Review · Reviewer_ryLS · 2022-10-20
**A new method for subgroup discovery, misrepresented title and contributions**

**Overall Rating:** 6

**Summary:**

The paper presents a new method for discovery of unknown subgroups with anomalous performance based on MDScan. Compared to prior methods, it supports defining subgroups as OR predicates. In an empirical evaluation, the method is able to find some groups not discovered by the prior approaches on three datasets.

**Strengths:**

- MDScan seems like a good fit for the task
- Ability to capture groups defined using OR predicates
- The proposed method is able to discover new groups with anomalous performance not found by other methods.

**Weaknesses:**

- The title and contributions seem misrepresented. Discovery of unknown groups with respect to which a model exhibits disparate performance or other formal notion of unfairness is a growing body of literature. "We discuss the notion of divergence of a subgroup that automatically balances the size and significance of systematic deviations". Other methods do consider statistical testing, thus this is not concrete enough distillations of the contributions. The title "Beyond Protected Attributes: Disciplined Detection of Systematic Deviations in Data" also sounds like an initial work on the problem which it is not.
- The solution to the multiple testing problem does not appear standard. E.g., Benjamini-Hochberg correction would be implemented in standard statistical packages. It is unclear why the proposed method was chosen rather than a simpler one.

Writing:
- The paper would benefit from a dedicated explanation of why OR predicate is valuable. E.g., what does discovery of a group with an OR provide compared to simply the discovery of two different groups.
- Explanation of the intuitions behind Equation 1 would also be clearer if it started with its likelihood ratio origin.

**Overall Recommendation:**

Accept. The paper would likely benefit from the workshop, and contribute to the topics of the workshop.

**Review Confidence:**

3: The reviewer is fairly confident that the evaluation is correct

---

### Official Review · Reviewer_Z5LR · 2022-10-21
**AutoStrat**

**Overall Rating:** 4

**Summary:**

The paper proposes a new algorithm, Automatic Stratification (AutoStrat), to automatically discover systematic deviations in data more trustworthy and responsible way by going beyond protected subgroups. The new algorithm allows a person to search not just over conjunctions (logical ANDS) of attributes but also over logical ORs of attributes. The authors use the method on three provided datasets to show some exciting deviations that earlier methods could not have found.

**Strengths:**

- The paper deals with an important problem, and the provided solution could be of great interest to the larger ML community.


**Weaknesses:**

- The paper reads more like a survey paper than a research paper. Although the discrepancy score is mentioned, it is not properly described or motivated.
- The paper makes many interesting claims but does not provide much justification for them. Given the current state of the paper, it is not possible to properly determine the validity of the method.

**Overall Recommendation:**

Although the paper deals with an important problem, the current state of the paper is not ready for publication. I would recommend giving a more involved discussion of the methodology along with rigorous proof of the validity of the proposed statistical tests.

**Review Confidence:**

2: The reviewer is willing to defend the evaluation, but it is quite likely that the reviewer did not understand central parts of the paper

---

### Decision · Program_Chairs · 2022-10-23

**Decision:**

Accept

**Comment:**

The paper studies the difficult problem of finding subgroups with potential bias, which is very important for fair ML. Reviewers have concerns on clarity of the paper and we hope the authors can address these concerns in the final revision.